# Inverse Salt Sensitivity of Blood Pressure Is Associated with an Increased Renin-Angiotensin System Activity

**John J. Gildea** [1], **Peng Xu** [1], **Katie A. Schiermeyer** [1], **Wei Yue** [1], **Robert M. Carey** [2], **Pedro A. Jose** [3,4] **and Robin A. Felder** [1,*]

[1] Department of Pathology, The University of Virginia, Charlottesville, VA 22903, USA
[2] Division of Endocrinology and Metabolism, Department of Medicine, The University of Virginia, Charlottesville, VA 22903, USA
[3] Division of Renal Diseases & Hypertension, Department of Medicine, School of Medicine and Health Sciences, The George Washington University, Washington, DC 20052, USA
[4] Department of Physiology/Pharmacology, School of Medicine and Health Sciences, The George Washington University, Washington, DC 20052, USA
* Correspondence: raf7k@virginia.edu; Tel.: +1-434-924-5151

**Abstract:** High and low sodium diets are associated with increased blood pressure and cardiovascular morbidity and mortality. The paradoxical response of elevated BP in low salt diets, aka inverse salt sensitivity (ISS), is an understudied vulnerable 11% of the adult population with yet undiscovered etiology. A linear relationship between the number of single nucleotide polymorphisms (SNPs) in the dopamine $D_2$ receptor (*DRD2*, rs6276 and 6277), and the sodium myo-inositol cotransporter 2 (*SLC5A11*, rs11074656), as well as decreased expression of these two genes in urine-derived renal proximal tubule cells (uRPTCs) isolated from clinical study participants suggest involvement of these cells in ISS. Insight into this newly discovered paradoxical response to sodium is found by incubating cells in low sodium (LS) conditions that unveil cell physiologic differences that are then reversed by mir-485-5p miRNA blocker transfection and bypassing the genetic defect by *DRD2* re-expression. The renin-angiotensin system (RAS) is an important counter-regulatory mechanism to prevent hyponatremia under LS conditions. Oversensitive RAS under LS conditions could partially explain the increased mortality in ISS. Angiotensin-II (AngII, 10 nmol/L) increased sodium transport in uRPTCs to a greater extent in individuals with ISS than SR. Downstream signaling of AngII is verified by identifying lowered expression of nuclear factor erythroid 2-related factor 2 (NRF2), CCCTC-binding factor (CTCF), and manganese-dependent mitochondrial superoxide dismutase (SOD2) only in ISS-derived uRPTCs and not SR-derived uRPTCs when incubated in LS conditions. We conclude that *DRD2* and *SLC5A11* variants in ISS may cause an increased low sodium sensitivity to AngII and renal sodium reabsorption which can contribute to inverse salt-sensitive hypertension.

**Keywords:** angiotensin; dopamine; dopamine receptor; inverse salt sensitivity; salt resistance; saltsensitivity

## 1. Introduction

Reducing dietary sodium (as NaCl) intake is one of the best proved non-pharmacological interventions for the prevention and treatment of hypertension [1–3]. Much is known about the molecular mechanisms involved in salt sensitivity (SS) since many studies have demonstrated a sustained increase in blood pressure (BP) with an increase in salt consumption [1–15]. Cardiovascular mortality is higher in individuals consuming high salt diet in salt-sensitive hypertensive and salt-sensitive normotensive than salt-resistant individuals [13–19]. However, little is known about individuals at the opposite end of the SS spectrum, i.e., inverse salt sensitivity (ISS) [20]. What is known is that death from cardiovascular diseases such as stroke, myocardial infarction, and congestive heart failure are also increased in some individuals who consume low amounts of sodium

(<1.2–2.5 g/day) [14,17,18,21–25]. A previously unanswered question is why should low salt intake lead to an increase in cardiovascular mortality? [14–18,21–25]. Others have measured the consequences (e.g., cardiovascular disease and death) in individuals with ISS. Long-term clinical studies (median 16 years, IQR 12–17 years [26,27]) showed that individuals with the ISS phenotype had a 15-fold greater risk of target organ damage than individuals with the salt resistance (SR) phenotype, as determined by clinical evaluation of the heart, kidney, vasculature, and brain, as well as a history of cardiovascular events similar to those seen in salt-sensitive individuals [26]. Our studies are the first to examine the biological, genetic, and epigenetic mechanisms associated with the paradoxical increase in BP in adults consuming a low sodium diet [4]. When approached from a mechanistic viewpoint, future studies will be better equipped to stratify cohorts with ISS appropriately determine if clinical outcomes can be related to known mechanisms of BP regulation in these individuals.

The dietary sodium recommendation by the Institute of Medicine/American College of Cardiology (ACC)/American Heart Association (AHA) [1,19] is intended as a "one size fits all" recommendation. However, it is becoming clear that each individual is genetically programmed with a "personal index of salt sensitivity" [28,29]. Currently, the guidelines set the upper limits but not the lower limits of NaCl intake to prevent and treat hypertension. As aforementioned, cardiovascular risk is increased above and below the intake of 2.5 to 6.0 g sodium/day [5,14,17,18,30–32]. Thus, more research is needed to understand the J-curve effect of salt intake on cardiovascular disease and death, especially since both ISS and SS constitute 11% and 18% of the general population, respectively, depending on cohort stratification [22–24,30].

In a study of 465 Pennsylvania Amish by Montasser et al. [23], a high sodium (Na) diet (280 mmol/d = 6440 mg/d) for 6 days, and then after a 6- to 14-day washout period, a standardized low-Na diet (40 mmol/d = 920 mg/d) was consumed for 6 days. The potassium (K) in the diet was held constant at 140 mmol/d in both diets. SS was determined by an increase in ambulatory BP [$\geq$5 mm Hg mean arterial pressure (MAP) or systolic BP $\geq$ 4 mm Hg] on high Na diet. In the studies of Overlack et al., the effects of 1 week of low-salt (20 mmol sodium/d) and 1 week of high-salt (300 mmol sodium/d) diet were investigated in a single-blind randomized study in 163 white non-obese normotensive subjects. Delta BP $\geq$ 4 mm Hg was use as a cutoff between SR and SS [24,25]. Castiglioni used a low sodium (30 mmol NaCl per day) and a high-sodium (200 mmol NaCl per day) diet, each for 5 days, in random order in 71 Italian participants [32]. The authors of all four studies observed a population of study participants that experienced a change in BP that was opposite to that of individuals with SS [23–25,32], that others have shown to be associated with increased mortality and morbidity [17,26].

Urinary surrogate markers, including renal proximal tubule cells (RPTCs), exosomes, and miRNA, hold promise for cost-effective methods to screen for SS and ISS [5,20,28,33–35]. Some genetic determinants of SS have been reported, as well [4,28,36–38]. Variants of the G protein-coupled receptor kinase type 4 (GRK4) have been shown to be associated with SR and SS in humans [4,37,39–41], and cause SR [42] or SS [43] when these human genes are expressed in mice. The genetic background of the mice and the variant expressed determine the resulting BP [44,45]. Variants in the human *SLC4A5* gene that encodes the sodium bicarbonate electrogenic cotransporter type 2 (NBCe2) are strongly associated with SS [4]. Angiotensin (Ang) type 1 receptor (AT$_1$R) is an antinatriuretic G protein-coupled receptor previously shown to increase calcium signaling via Ang II in ISS [20]. Other components of the renin-angiotensin system (RAS) may also play a role in the pathogenesis of ISS, such as aminopeptidase N (APN) which converts Ang III into angiotensin IV (Ang IV), reducing the levels of Ang III, the preferred endogenous agonist for the natriuretic Ang II type 2 receptor (AT$_2$R), resulting in sodium retention and elevated BP [46,47]. APN has variable effects on BP [48–50]. Decreased cerebral APN activity increases Ang III levels and BP; the cerebral and paraventricular nucleus infusion of APN decreases BP [49]. However, in the kidney, a defect in Ang III signaling through the AT$_2$R occurs in both prehypertensive and

hypertensive spontaneously hypertensive rats (SHR) [51]; increased production of Ang III and decreased renal APN activity rescues this defect and inhibits renal sodium transport and decreases BP in SHR [50]. By contrast, nanomolar concentrations of Ang IV have been reported to increase BP in rats by interacting with the $AT_1R$ [48]. Some effects of Ang IV in the kidney may be through the binding and inhibition of insulin-regulated aminopeptidase (IRAP) activity [52]; global IRAP knockout mice have increased vasopressin sensitivity, yet no change in blood pressure [53].

The dopamine type 2 receptor ($D_2R$) negatively interacts with the $AT_1R$ in many tissues, including the kidney [54,55] *DRD2* rs6276 (1347G>A) at 3′UTR (MAF 0.484) has decreased $D_2R$ function and *DRD2* rs6277 (957C>T exon7 synonymous mutation, MAF = 0.273) has decreased *DRD2* mRNA stability [56]. $D_2R$ has a protective role against inflammation and these *DRD2* variants increase inflammation and fibrosis in human and mouse RPTCs [57–60]. The present studies tested the hypothesis that *DRD2* variant-mediated dysregulation of the RAS contributes to the pathogenesis of ISS. Other genes that are expressed in the kidney, such as SMIT1 (*SLC5A3*) [61] and SMIT2 (*SLC5A11*, *SGLT6*) ($Na^+$/myoinositol cotransporters), may be related to ISS. SMIT2 is responsible for the apical sodium/myoinositol transport in the RPT [62].

Although the harmful effects of lowering sodium consumption is controversial for the general population that is not hypertensive or salt-sensitive, the simple observation that there are individuals in every SS study that have a paradoxical blood pressure response to lowering sodium consumption is not. Our group has previously demonstrated direct ex vivo cell physiologic differences in urine-derived renal proximal tubule cells (uRPTCs) in ISS individuals. We extend these studies by culturing these uRPTCs to show that they have enduring sodium-induced phenotypic changes that are amenable to genetic and epigenetic phenotypic reversal centered on $D_2R$ expression and Ang II sensitivity.

## 2. Materials and Methods

### 2.1. Living Renal Proximal Tubules Obtained from Spot Urines

The human tissues and urine specimens used in our studies were obtained in accordance with a University of Virginia Institutional Review Board-approved protocol that adheres to the Declaration of Helsinki and the most recent version of the USA Code of Federal Regulations Title 45, Part 46 [33]. We obtained spot urine collections from salt study participants described in a previous publication [4]. Briefly, we used the statistical tests to classify participants who successfully completed our salt diets (based on 24-h urine salt balance evaluation) into ISS, SR, and SS. Statistical justification for the cutoff points between the ISS, SR, and SS groups was performed, as previously published [47].

### 2.2. uRPTC Culture

Spot, mid-stream and clean-catch, urine (>200 mL) was centrifuged within 2 h after collection in 50 mL tubes at 1000 relative centrifugal force (RCF) for 3 min. The supernatant was removed, and the pellets were combined into one 50 mL tube, filled with PBS (Dulbecco's PBS + $CaCl_2$ + $MgCl_2$, D8662, Sigma-Aldrich, St. Louis, MO, USA), centrifuged again, and the supernatant removed. Two ml of culture medium (REGM, cc-3190, Walkersville, MD, USA) were added to the pellet and plated in a 96-well tissue culture plate at 200 µL per well. Fifty µL of spent media were exchanged with fresh media every other day. The cell colonies appeared in 7–14 days. The cells were immortalized with lentiviral hTERT, as previously described [63].

### 2.3. Plasma Membrane $D_1R$ and $AT_2R$ Expression

uRPTCs were directly isolated and imaged for $D_1R$ [20], and for $AT_1R$ [64], as previously described, with the exception that a monoclonal antibody was generated related with the prospects of developing diagnostic clinical assays. Briefly, in TERT-immortalized uRPTCs, intracellular sodium was increased by the ionophore monensin [65] (MON, 10 µmol/L, 1 h), and $D_1R$ and $AT_2R$ recruitments were measured directly, using fluo-

rescently labeled antibodies in collagen-coated glass-bottomed 96-well plates. The effect of the D1-like receptor agonist fenoldopam (FEN, 1 $\mu$mol/L, 30 min)-induced plasma membrane $AT_2R$ recruitment was performed identically, as in the $D_1R$ assay [20], except for the additional use of $AT_2R$ monoclonal antibodies.

### 2.4. Plasma Membrane $D_2R$ and APN Expression

Fixed, non-permeabilized in-cell western blotting, using directly labeled primary antibody against a synthetic (PLKEAARRC) peptide, corresponding to amino acids 239–246 of the third extracellular loop of the human $D_2R$ (324393, EMD Millipore, Burlington, MA, USA) was performed, as previously described [58–60,66]. Briefly, uRPTCs, plated in collagen-coated 96-well glass bottom microplates, were fixed with 4% paraformaldehyde without detergent in phosphate-buffered saline (PBS), blocked with Odyssey blocking reagent (927–40003, Licor, Lincoln, NE, USA) in PBS, and incubated with an extracellular epitope-specific rabbit anti-$D_2R$ antibody (EMD Millipore), directly conjugated with Alexa-647 (ab269823, Abcam, Cambridge, MA, USA), or purified monoclonal antibody against APN, also known as CD13, clone 452 (provided by Dr. Meenhard Herlyn, The Wistar Institute of Anatomy and Biology, Philadelphia, PA, USA), directly conjugated with Alexa-647, and read in a fluorescent microplate reader in well-mode (PHERAstar FS, BMG, Raleigh, NC, USA). Plating efficiency was normalized by taking the in-cell $D_2R$ fluorescence values, divided by simultaneous measurement of Hoechst 33,342 (735969, 1:2000, Thermo Fisher, Waltham, MA, USA) staining.

### 2.5. Total Cellular CTCF, NRF2, and SOD2 Expression

Fixed-permeabilized in-cell western blotting, using directly labeled primary antibody against CCCTC-binding factor (CTCF, sc-271474, Santa Cruz Biotechnology, Dallas, TX, USA), nuclear factor erythroid 2-related factor 2 (NRF2, sc-365949, Santa Cruz Biotechnology), and manganese-dependent mitochondrial superoxide dismutase (SOD2, sc-137254, Santa Cruz Biotechnology) was performed as above, with the exception that the cells were permeabilized with 0.2% Triton-X100 for 5 min, following fixation.

### 2.6. $D_2R$ Binding Assay

$D_2R$ fluorescence binding was measured by treating live cells with a fluorescently labeled $D_2R$ antagonist, spiperone (sc-206629, Santa Cruz Biotechnology) [67–69], or bodipy-propanoic acid N-phenethylspiperone amide (aka Spiperone-FL, 100 nmol/L, 2 h). The cells were washed, and the fluorescence was read in a fluorescent plate reader (ex 488 em 510). Spiperone has similarly high affinities for two members of the dopamine $D_2$-like receptor family, $D_2R$ and $D_4R$, and 10-fold lower affinity for the $D_3R$ and many-fold lower affinity for members of the dopamine $D_1$-like receptor family, $D_1R$, and $D_5R$ [67,68]. However, addition of a $D_4R$ selective antagonist (L-741742, 200 nmol/L), or a $D_3R$ antagonist (U99194A, 1 $\mu$mol/L) [69–71] did not significantly decrease spiperone-FL binding (data not shown).

### 2.7. Dopamine $D_2R$ siRNA

uRPTCs were transfected overnight with 50 nmol/L *DRD2* siRNA or control non-targeting siRNA (D-001210-02-50, Dharmacon, Cambridge, UK), using Lipofectamine RNAiMAX (13778075, Thermo Fisher). $D_2R$ binding was measured 48 h later.

### 2.8. $D_2R$ Overexpression

We purchased a lentiviral construct (*DRD2* long) from GeneCopoeia. The plasmid was packaged into virus with compatible packaging plasmids, using HEK-293T cells (Clontech Laboratories, Mountain View, CA, USA). The lentivirus was added to uRPTCs at 30–40% confluence for 18–20 h, then removed and replaced with regular growth medium. After 48 h, the medium was changed to selection medium, containing puromycin (Sigma, 5 $\mu$g/mL). The *DRD2* long was used, instead of *DRD2* short, because the former is the isoform expressed in the kidney [71].

### 2.9. Sodium Influx Assay

Details and specificity of this assay, in which ouabain was used to inhibit Na$^+$-K$^+$/ATPase activity, permitting the quantification of sodium accumulation in RPTCs, were previously published [72]. Briefly, the uRPTCs were cultured in 96-well glass bottom collagen-coated Matrical plates (Dot Scientific, Burton, MI, USA, MGB096-1-2LGL) at 37 °C until they reached 80% confluence. The cells were serum-starved overnight prior to loading with the sodium ion indicator, sodium-binding benzofuran isophthalate acetoxymethyl ester (SBFI, 5 µmol/L; ThermoFisher), with 0.04% Pluronic F-127 for 2 h in PBS with calcium (0.9 mmol/L) and magnesium (0.49 mmol/L). The cells were washed twice and allowed to recover at 37 °C in serum-free media for 30 min. They were washed two more times with PBS and incubated at room temperature. The cells were then placed in a fluorescence plate reader (Pherastar FS, BMG) and ratiometric readings (340 nm excitation and 510 nm emission/380 nm excitation and 510 nm emission) were recorded every 30 s for 30 min. A 5-min baseline reading was acquired prior to automatic injection of ouabain (100 µmol/L), to inhibit basolateral sodium transport. The change in sodium accumulation was measured as the change in 340/380 ratio at the 30-min time point minus the 5-min average reading.

### 2.10. Seminaphtharhodafluor (pH-Sensitive Dye, SNARF), Sodium-Hydrogen Exchanger Type 3 (NHE3), and Sodium-Dependent pH Recovery Assay in Low Sodium Concentrations

uRPTCs, plated in thin bottom plastic tissue culture-treated 96 well plates (uClear, Greiner Bio-One, Stonehouse, UK) were incubated in low sodium concentration (90 mmol/L sodium) for 2 h, using serum-free DMEM/F12 media diluted with water and mannitol to make the solution iso-osmolar (290 mOsm/L). In order to control for other factors in the physiologic serum-free media, normal serum-free media in DMEM/F12 was also diluted with the same amount of water as low sodium media but adjusted to 140 mmol/L sodium as sodium chloride. uRPTCs were loaded with carboxy SNARF-1 AM acetoxymethyl ester acetate in low-sodium Hank's Balanced Salt Solution (HBSS, 90 mmol/L sodium, osmolality adjusted to 290 mOsm/L with mannitol) for 1 h, washed and incubated with 20 mmol/L ammonium chloride for 10 min, and changed to serum-free HEPES-buffered saline (HBS, 5 mmol/L HEPES, 140 mmol/L choline chloride, 1 mmol/L calcium chloride, 1 mmol/L magnesium chloride, 4 mmol/L potassium chloride), to stabilize the extracellular pH in the absence of $CO_2$, but without bicarbonate which would lower the intracellular pH, which is necessary in order to measure sodium-dependent pH recovery, without interference with NBCe2 activity [70]. The microplate was transferred to a fluorescence microplate reader (PHERAstar FS, BMG), equipped with a single excitation (520 nm) and dual emission (640 nm/580 nm) custom filter set for SNARF-1 and baseline reading taken for 5 min before switching to low sodium containing HBS (5 mmol/L HEPES, 90 mmol/L sodium chloride, 50 mmol/L choline chloride, 1 mmol/L calcium chloride, 1 mmol/L magnesium chloride, 4 mmol/L potassium chloride). Full plate measurements were taken every 10 s and the slope of the first 2 min was used to record pH recovery. The pH was internally calibrated using the nigericin high potassium calibration method [73].

### 2.11. Micro Ribonucleic Acid 485-5p (miR-485-5p) Mimic and Blocker Transfection

miR-485-5p mimic (50 nmol/L), miR-485-5p blocker [74–76], or control miRNA (Mission, Sigma-Aldrich) was transfected overnight into the uRPTCs, using Lipofectamine RNAiMAX™ and D$_2$R binding measured after 48 h.

### 2.12. SLC5A11 SNP and qRT-PCR

*SLC5A11* SNP rs11074656 (C/T), *DRD2* SNPs rs6276 and rs6277 were identified using Taqman PCR probes (Thermo Fisher), a two-step fast cycling real-time PCR machine (BioRad CFX96 Connect, Hercules, CA, USA), and SNP determination by CFX Maestro 2. *SLC5A11* mRNA was quantified using real-time quantitative reverse transcriptase PCR (qRT-PCR), as previous described [77]. Primers: *SLC5A11* sense 5′-GCCTCCACAGTTA

GATCCCC-3′; *SLC5A11* anti-sense 5′-CAGAACTAGCACCGCGATG-3′; actin sense 5′-AGAAAATCTGGCACCACACC-3′; Actin anti-sense 5′-CTCCTTAATGTCACGCACGA-3′.

### 2.13. Statistical Analysis

Statistical analysis was performed using GraphPad Prism 9 software (GraphPad, San Diego, CA, USA) as well as engaging the University of Virginia's Medical School's Department of Biomedical Statistics. Response differences were evaluated using t-test and analysis of variance with post hoc Bonferroni and Tukey tests for repeated measures. For most studies, we used power calculations based on the endpoints of major interest: sodium transport, expression, and activity of the members of the RAS, $D_2R$ expression and activity, miRNA expression, and downstream messengers. We justified selecting a Delta MAP of $\geq +7$ mm Hg for Defining SS and $\leq -7$ mm Hg for ISS (With the assistance of our collaborating biostatistician our statistical method that assures us that the cutoffs for classifying ISS and SR are no longer arbitrary [7].

### 3. Results

#### 3.1. $D_1R$ and $AT_2R$ Plasma Membrane Recruitment

With 1-h 10 μM monensin (MON) treatment, more $D_1Rs$ (Figure 1A) and $AT_2Rs$ (Figure 1B) are recruited to the plasma membrane in cells from individuals with ISS, but not in cells from individuals with SR ($D_1R$: MON/VEH: SR, $1.022 \pm 0.038$, $n = 6$; ISS, $1.635 \pm 0.178$, $n = 6$; t-test, $p < 0.01$; $AT_2R$: MON/VEH: SR, $0.965 \pm 0.04$, $n = 6$; ISS, $1.434 \pm 0.107$, $n = 6$; t-test, $p < 0.01$).

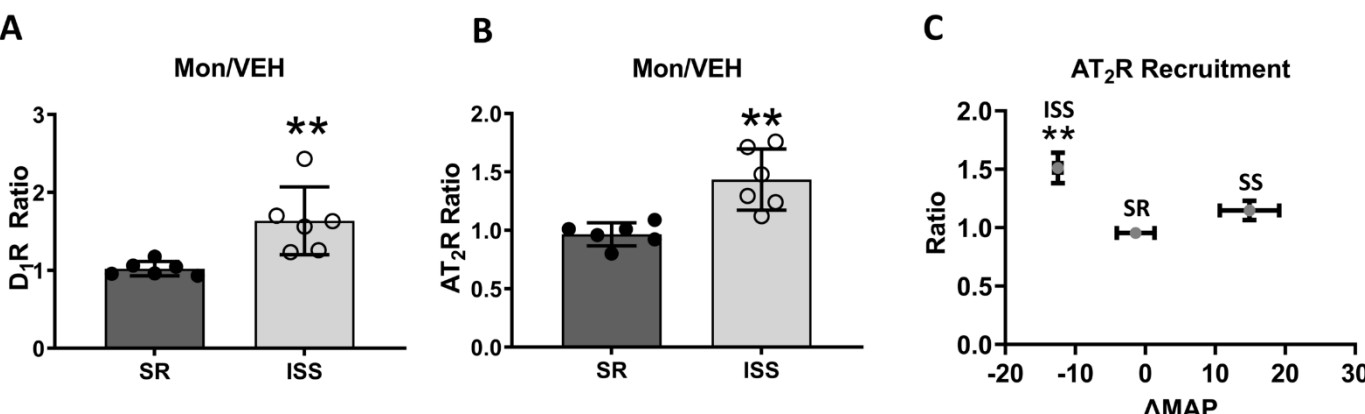

**Figure 1.** $D_1R$ and $AT_2R$ recruitment in urine-derived renal proximal tubule cells (uRPTCs) from salt-resistant (SR) and inverse salt-sensitive (ISS) subjects. (**A**,**B**) Under 1-h 10 μM monensin (MON) treatment, there are more $D_1Rs$ (**A**) and $AT_2Rs$ (**B**) recruited to the cell membrane in ISS uRPTCs, but not in SR uRPTCs ($D_1R$: MON/VEH: SR, $1.022 \pm 0.038$, $n = 6$; ISS, $1.635 \pm 0.178$, $n = 6$; t-test, ** $p < 0.01$; $AT_2R$: MON/VEH: SR, $0.965 \pm 0.04$, $n = 6$; ISS, $1.434 \pm 0.107$, $n = 6$; t-test, ** $p < 0.01$). (**C**) Correlation between the degrees of salt sensitivity (SS), expressed as a change in MAP (ΔMAP, High Salt–Low Salt) in response to increased dietary sodium and FEN-stimulated $AT_2R$ membrane recruitment in directly evaluated uRPTCs. The SS index of each SS class is depicted on the X axis expressed as the mean ΔMAP vs. $AT_2R$ 647/488 (representing $AT_2R$ recruitment from the cytosol to the plasma membrane). A higher ratio of FEN-induced plasma membrane $AT_2R$ expression is observed in ISS individuals compared with SR individuals ($AT_2R$: FEN/VEH: ISS, $1.51 \pm 0.13$, $n = 4$, mean ΔMAP $= -12.5$ mm Hg; SR, $0.955 \pm 0.04$, $n = 4$, mean ΔMAP $= -1.38$; SS, $1.148 \pm 0.082$, $n = 3$, mean ΔMAP $= +14.9$ mm Hg; one-way ANOVA, Holm-Sidak post hoc test ** $p < 0.01$).

#### 3.2. Direct $AT_2R$ Plasma Membrane Recruitment in uRPTC

We have previously shown that the sodium-induced recruitment of the $D_1R$ to human RPTC plasma membrane correlates with the degree of SS in a clinical study [20]. Here, we extend those studies by showing the relationship between $D_1$-like receptor agonist

FEN-induced $AT_2R$ plasma membrane recruitment and the degree of SS. Unlike the linear relationship previously found for $D_1R$ and SS [20], $AT_2R$ plasma membrane recruitment was only increased in subjects with ISS ($AT_2R$: FEN/VEH: ISS, $1.511 \pm 0.13$, $n = 4$; SR, $0.955 \pm 0.04$, $n = 4$; SS, $1.148 \pm 0.082$, $n = 3$; one-way ANOVA, Holm-Sidak post hoc test ** $p < 0.01$) (Figure 1C).

### 3.3. $D_2R$ Binding

Immunostaining showed that $D_2R$ was expressed in the human RPT (Figure 2A). To determine if $D_2R$ participates in the pathogenesis of ISS, ten uRPTCs, 5 isolated from clinical study participants with ISS and 5 isolated from clinical participants with SR, were grown, isolated, immortalized with Tert, and plated as 8 technical replicates in 96-well glass-bottom collagen-coated Matrical plates and grown to 80% confluence. Plasma membrane $D_2R$ expression was measured using an extracellular epitope-specific rabbit anti-$D_2R$ antibody and found to be reduced by $25.3 \pm 0.5\%$ in uRPTCs from clinical study participants with ISS ($p < 0.01$, $n = 54$/group), relative to SR cells (Figure 2B,C).

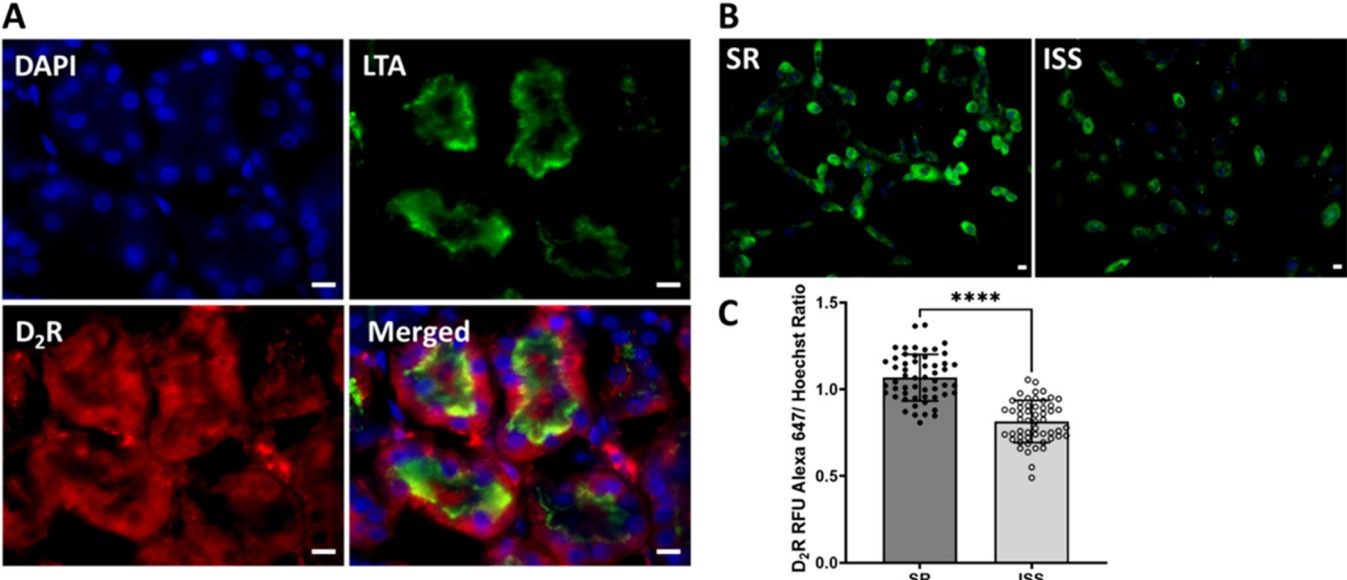

**Figure 2.** $D_2R$ immunostaining in human kidney and renal proximal tubule cells. (**A**) $D_2R$ is stained red in the kidney. LTA, *Lotus tetragonolobus*, a proximal tubule marker. (**B**) Plasma membrane expression of $D_2R$ in renal proximal tubule cells. Nuclei are stained blue. Scale bar = 10 μm (**C**) Plasma membrane $D_2R$ expression, using an extracellular epitope-specific anti-$D_2R$ polyclonal antibody directly labeled with Alexa 647, is lower in fixed uRPTCs from individuals with ISS than individuals with SR (SR, $1.07 \pm 0.02$; vs. ISS, $0.81 \pm 0.02$; **** $p < 0.001$, $n = 54$; *t*-test). ISS = inverse salt-sensitive; SR = salt-resistant.

### 3.4. Total Cellular CTCF, NRF2, and SOD2 Expression

Since *DRD2* contains five topologically associated domains containing CCCTC-binding factor (CTCF) studied by Chipseq [78], we measured CTCF expression in uRPTCs. CTCF was reduced by $16.75 \pm 1.22\%$ in ISS uRPTCs incubated in low salt concentration (90 mmol/L sodium) ($p < 0.01$ ISS 90 mmol/L sodium vs. ISS 140 mmol/L (normal) or 190 mmol/L sodium as NaCl (high) salt concentration, $n = 9$) (Figure 3A). Because $D_2Rs$ and dopamine are implicated in redox signaling in the RPT, we measured the expression of the protein responsible for controlling the majority of antioxidant response elements [79,80], i.e., basic leucine zipper (bZIP) protein (NRF2, Nuclear Factor Erythroid 2 Like 2, aka NFE2L2). NRF2 expression in uRPTCs was reduced by $12.73 \pm 0.49\%$ in ISS only under low salt (90 mmol/L) concentration ($p = 0.0032$ ISS low salt vs. ISS 140 or 190 mmol/L salt, as NaCl, $n = 18$) (Figure 3B). To validate the functionality of the reduced expression of NRF2,

superoxide dismutase 2 (SOD2), a downstream antioxidant gene regulated by NRF2 [81], was studied. SOD2 expression in uRPTCs was reduced by $34.23 \pm 6.52\%$ in ISS only under low NaCl concentration ($p < 0.01$ ISS low salt vs. ISS 140 or 190 mmol/L NaCl, $n = 18$) (Figure 3C).

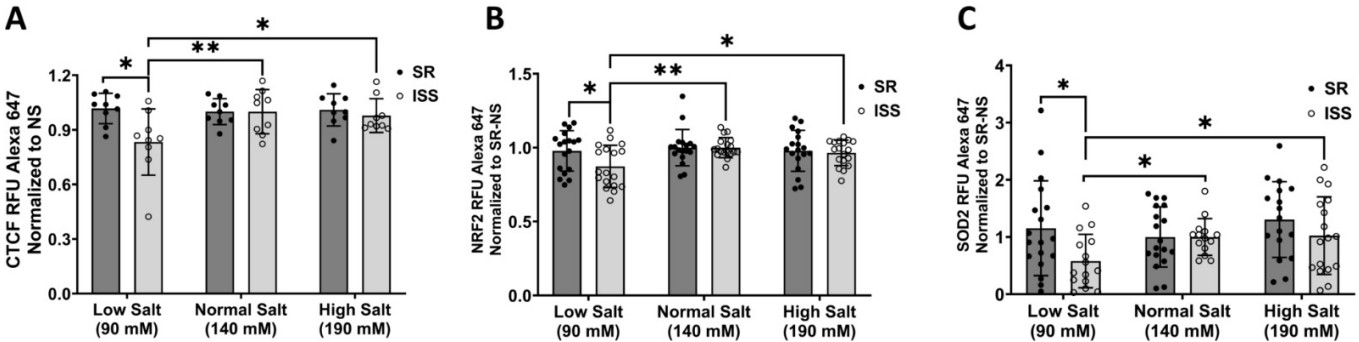

**Figure 3.** CTCF, NRF2 and SOD expression under salt treatment by In-Cell Western. SR and ISS cells were treated with low salt (90 mM), normal salt (140 mM) and high salt (190 mM) overnight. (**A**) CCCTC-binding factor (CTCF) expression decreased only in uRPTCs from ISS subjects incubated under low salt concentration (SR low salt, $1.02 \pm 0.03$ vs. ISS low salt, $0.83 \pm 0.06$; ISS low salt vs. ISS normal salt, $1 \pm 0.04$; ISS low salt vs. ISS high salt, $0.98 \pm 0.03$, * $p < 0.05$, ** $p < 0.01$, $n = 9$/group, two-way ANOVA, Holm-Sidak post hoc test). (**B**) Nuclear factor erythroid 2-related factor 2 (NRF2) expression significantly decreased only in ISS uRPTCs under low salt concentration (SR low salt, $0.98 \pm 0.03$; vs. ISS low salt, $0.87 \pm 0.03$; * $p < 0.05$, $n = 18$/group; ISS low salt vs. ISS normal salt, $1 \pm 0.02$; ISS low salt vs. ISS high salt, $0.96 \pm 0.02$, * $p < 0.05$, ** $p < 0.01$, $n = 18$/group, Two-way ANOVA, Holm-Sidak post hoc test). (**C**) SOD2 was significantly decreased only in ISS uRPTCs under low salt concentration (SR low salt, $1.15 \pm 0.19$; vs. ISS low salt, $0.58 \pm 0.11$; ISS low salt vs. normal salt $1 \pm 0.08$; ISS low salt vs. ISS high salt $1.02 \pm 0.16$, * $p < 0.05$, $n = 18$/group, ISS low salt vs. ISS normal salt, one-way ANOVA, Holm-Sidak post hoc test). ISS = inverse salt-sensitive; SR = salt-resistant.

### 3.5. Ang II-Stimulated Sodium Influx in ISS and SR

Our previous study showed that uRPTCs from clinical study participants with ISS fed a low salt diet (10 mEq/day) for one week that had an increase in BP, displayed a heightened Ang II-mediated increase in intracellular calcium, relative to individuals with SR or SS [20]. Therefore, Ang II sensitivity was evaluated in these uRPTCs. We measured the effect of Ang II (10 nmol/L) on sodium influx with the sodium-sensitive ion indicator, SBFI. The uRPTCs were incubated with vehicle or Ang II (10 nmol/L, 10 min), in which basolateral sodium outflux was inhibited by incubating cells in media with ouabain (100 μmol/L). Sodium influx was similar in vehicle-treated uRPTCs from individuals with SR and ISS. uRPTCs showed a robust increase in sodium influx when stimulated with Ang II in SR ($15,426 \pm 900$ RFU SR vehicle (VEH) vs. $20,357 \pm 667$ RFU SR Ang II, $n = 15$, * $p < 0.01$) and ISS ($15,571 \pm 904$ RFU ISS VEH vs. $26,942 \pm 815$ RFU ISS Ang II, $n = 15$, ‡ $p < 0.001$) individuals (Figure 4). However, the sodium influx in response to Ang II was greater in uRPTCs from individuals with ISS than individuals with SR (ISS $26942 \pm 815$ RFU vs. SR $20357 \pm 667$ RFU, $n = 15$, † $p < 0.001$) (Figure 4).

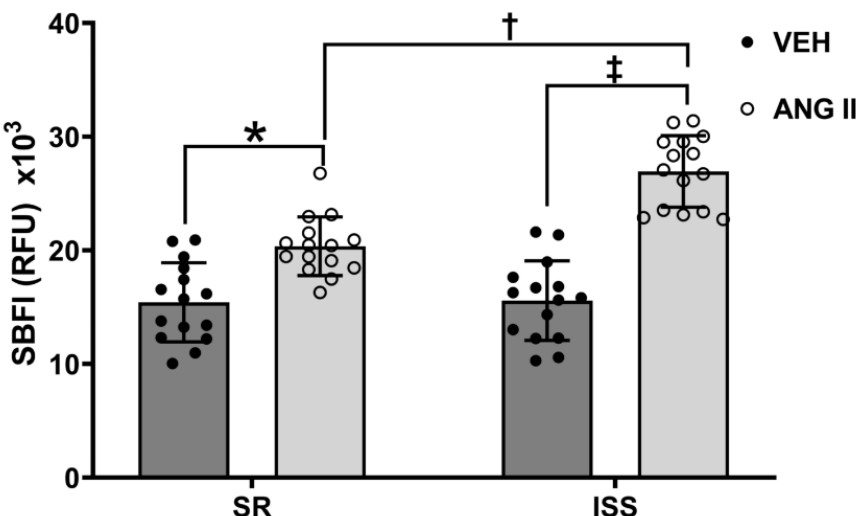

**Figure 4.** Sodium influx in SR and ISS cells under ANG II treatment. Ang II increases luminal sodium transport to a greater extent in uRPTCs from individuals with ISS than those from individuals with SR. uRPTCs, incubated in low salt concentration (90 mM Sodium as NaCl, osmolality adjusted to 290 mOsm/L with mannitol) for 2 h, loaded with sodium-sensitive ion indicator SBFI (Ex 340/Em 510), in the presence of ouabain to inhibit basolateral sodium exit, were studied with and without 10 nmol/L Ang II. There is a robust increase in sodium influx with Ang II stimulation in uRPTCs from both SR (15426 $\pm$ 900 RFU SR vehicle vs. 20,357 $\pm$ 667 RFU SR Ang II, $n = 15$, * $p < 0.01$) and ISS (15571 $\pm$ 904 RFU ISS vehicle vs. 26,942 $\pm$ 815 RFU ISS Ang II, $n = 15$, one-way ANOVA, Holm-Sidak test, $\ddagger$ $p < 0.001$) individuals. However, the Ang II-mediated increase in sodium influx is greater in individuals with ISS than individuals with SR (20357 $\pm$ 667 RFU SR Ang II vs. 26,942 $\pm$ 815 RFU ISS Ang II, $n = 15$, one-way ANOVA, Holm-Sidak test, $\dagger$ $p < 0.001$). ISS = inverse salt-sensitive; SR = salt-resistant.

### 3.6. D$_2$R Extracellular Epitope Binding and D$_2$R Overexpression in Living uRPTCs

To determine the role of D$_2$R in the differential effect of Ang II in the sodium influx in uRPTCs from individuals with SR or ISS, we decreased D$_2$R expression with D$_2$R siRNA or overexpressed D$_2$R and measured D$_2$-like receptor binding by live-96-well fluorometry, using fluorescent spiperone, a D$_2$-like receptor antagonist with high affinities to the members of the D$_2$-like receptor family, D$_2$R = D$_4$R = D$_3$R, [67,68]. In uRPTCs treated with control siRNA, spiperone binding was lesser in individuals with ISS than those with SR (36.9% $\pm$ 2.6% reduction in ISS control siRNA vs. SR control siRNA, $n = 5$, $\dagger$ $p < 0.01$) (Figure 5A), confirming the results using the extracellular epitope-specific D$_2$R antibody (Figure 2). Spiperone binding in SR uRPTCs was reduced by *DRD2* siRNA (37.1% $\pm$ 2.0% reduction in SR *DRD2* siRNA vs. SR control, $n = 5$, * $p < 0.01$). Spiperone binding in ISS uRPTCs was reduced by *DRD2* siRNA (42.7% $\pm$ 6.0% reduction in ISS *DRD2* siRNA vs. ISS control, $n = 5$, * $p < 0.01$). The percent reduction in spiperone binding was not different between the SR and ISS uRPTCs. D$_2$R overexpression increased spiperone binding in SR RPTCs (36.7% $\pm$ 7.9% increase over SR control siRNA ($n = 5$, $\ddagger$ $p < 0.01$). D$_2$R overexpression also increased spiperone binding in ISS uRPTCs (22.5% $\pm$ 4.8% increase over SR control siRNA ($n = 5$, $\ddagger$ $p < 0.01$). The percent increase in spiperone binding in D$_2$R overexpression in uRPTCs was not different between individuals with SR or ISS (Figure 5A). These results suggest that the spiperone binding assay is specific and sensitive enough to measure endogenous and exogenous D$_2$R expression in uRPTCs, and not affected by SS, ISS or otherwise.

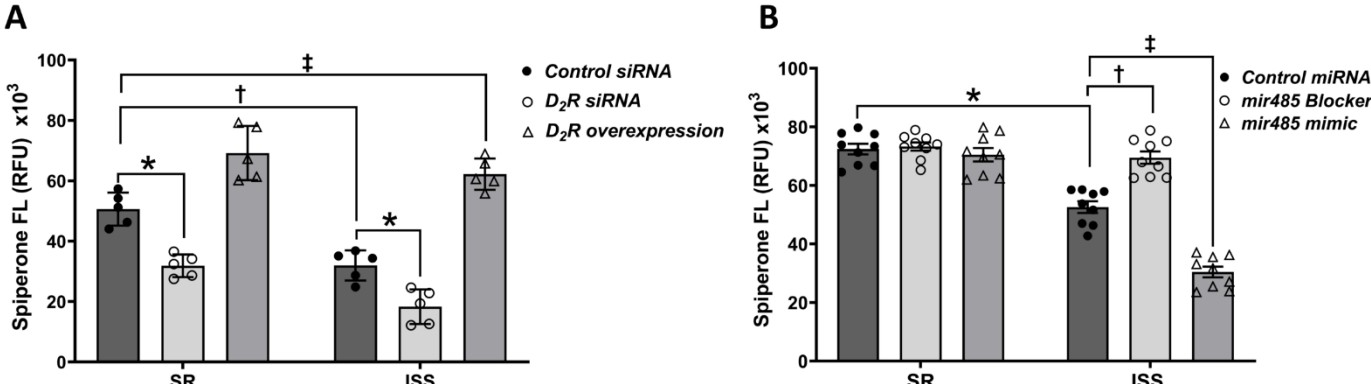

**Figure 5.** $D_2R$ cell surface expression when knocked down, overexpressed, or blocked in cells from subjects with SR or ISS. (**A**) siRNA decreases $D_2R$ cell surface expression in uRPTCs from both SR and ISS subjects (37.1% $\pm$ 2.0% reduction in SR *DRD2* siRNA vs. SR control, $n = 5$, and 42.7% $\pm$ 6.0% reduction in ISS *DRD2* siRNA vs. ISS control, $n = 5$, *, †, $p < 0.01$, Two-way ANOVA, Tukey's test). The overexpression of $D_2R$ is relatively greater in uRPTCs from ISS than SR (22.5% $\pm$ 0.9% increase for ISS $D_2R$ overexpression over SR control siRNA, $n = 5$, ‡ $p < 0.01$, Two-way ANOVA, Tukey's test). (**B**) mir-485-5p is involved in the reduced expression of the $D_2R$ in uRPTCs from individuals with ISS and *DRD2* rs6276 because mir-485 blocker normalizes *DRD2* expression in these cells (52619 $\pm$ 2001 RFU ISS control miRNA vs. 69,496 $\pm$ 2108 RFU ISS miRNA blocker and vs. 30,434 $\pm$ 1824 RFU ISS miRNA mimic, $n = 9$, Two-way ANOVA, Tukey's test, *, †, ‡, $p < 0.01$); there is no effect in uRPTCs from individuals with SR. Mir-485 mimic decreases spiperone binding in uRPTCs from individuals with ISS and *DRD2* rs6276; there is no effect in uRPTCs from individuals with SR. ISS = inverse salt-sensitive; SR = salt-resistant.

### 3.7. Epigenetic Regulation of the $D_2R$ by Mir-485-5p

We studied the effect of miR-485-5p mimic and blocker on spiperone ($D_2$-like receptor antagonist) binding to test the hypothesis that endogenous miRNA may regulate $D_2R$ expression in uRPTCs [67,68]. The *DRD2* SNP rs6726, located in the *DRD2* 3′UTR (untranslated region), results in a sequence that could cause an epigenetic downregulation of the synthesis of the $D_2R$ [72]. miR-9-5p, miR-485-5p, and miR-137 regulate schizophrenia risk genes and miR-9-5p targets the *DRD2*, a drug-target in schizophrenia [74–76]. An miR-485-5p mimic and blocker were synthesized and transfected into uRPTCs to increase or decrease miR-485-5p binding to target genes, respectively. Spiperone binding was lesser in control miRNA-treated uRPTCs from individuals with ISS than those with SR (Figure 5B), similar to that found in untreated uRPTCs (Figure 5A). miR-485-5p blocker increased while miR-485-5p mimic decreased the expression of the $D_2R$ only in uRPTCs from individuals with ISS and *DRD2* SNPs rs6276/rs6277 (52,619 $\pm$ 2001 RFU ISS control miRNA vs. 69,496 $\pm$ 2108 RFU ISS miRNA blocker or vs. 30,434 $\pm$ 1824 RFU ISS miRNA mimic, $n = 9$, †, ‡ $p < 0.01$) (Figure 5B). Thus, miR-485-5p only binds to the *DRD2* SNPs, leading to the reduction in $D_2R$ expression. The miRNA mimic behaves just like the endogenous miR-485-5p miRNA and reduced the already decreased expression of $D_2R$ in uRPTCs from individuals with ISS. The spiperone binding returned to normal when the cells were transfected with a miR-485-5p miRNA blocker which inhibited the endogenous miRNA. Because miR-485-5p has the highest energy binding level to *DRD2* rs6276 [74], the above result suggests that rs6276 creates a neomorphic genetic alteration since the miR-485-5p mimic and blocker have no significant effect on $D_2R$ wild-type. The lower $D_2R$ expression in ISS uRPTCs due to miR-485-5p binding at the rs6276 SNP site can be bypassed by stable overexpression (Figure 5A), since the $D_2R$ lentiviral construct has a different 3′UTR (sv40 3′UTR and poly A sequence), relative to the endogenous *DRD2* gene.

### 3.8. The Effect of $D_2R$ Overexpression on Sodium Transport in uRPTCs

Stable *DRD2 long* (D2L)-overexpressing uRPTCs were then evaluated to determine if the Ang II-stimulated sodium transport observed in uRPTCs from individuals with ISS when incubated in media with a low salt (NaCl) concentration, could be reverted to cells that respond to Ang II in a normal manner, similar to uRPTCs from individuals with SR. SNARF1 microplate fluorometry was utilized to measure NHE3-dependent pH recovery. Ang II increased pH recovery in low salt (90 mM) concentration only in ISS uRPTCs without *DRD2*-long (wo) overexpression (ISS Ang II vs. ISS VEH, $n = 5$ per group, * $p < 0.05$), an effect that was completely blocked by the $AT_1R$ antagonist losartan (LOS) (ISS Ang II vs. ISS Ang II LOS, $n = 5$/group, ‡ $p < 0.05$) (Figure 6A), which by itself had no effect. Stable overexpression of *DRD2 long* (w D2L) completely blocked the low salt Ang II-dependent increase in pH in ISS uRPTCs (ISS Ang II vs. ISS Ang II w D2L, $n = 5$/group, † $p < 0.05$) (Figure 6A).

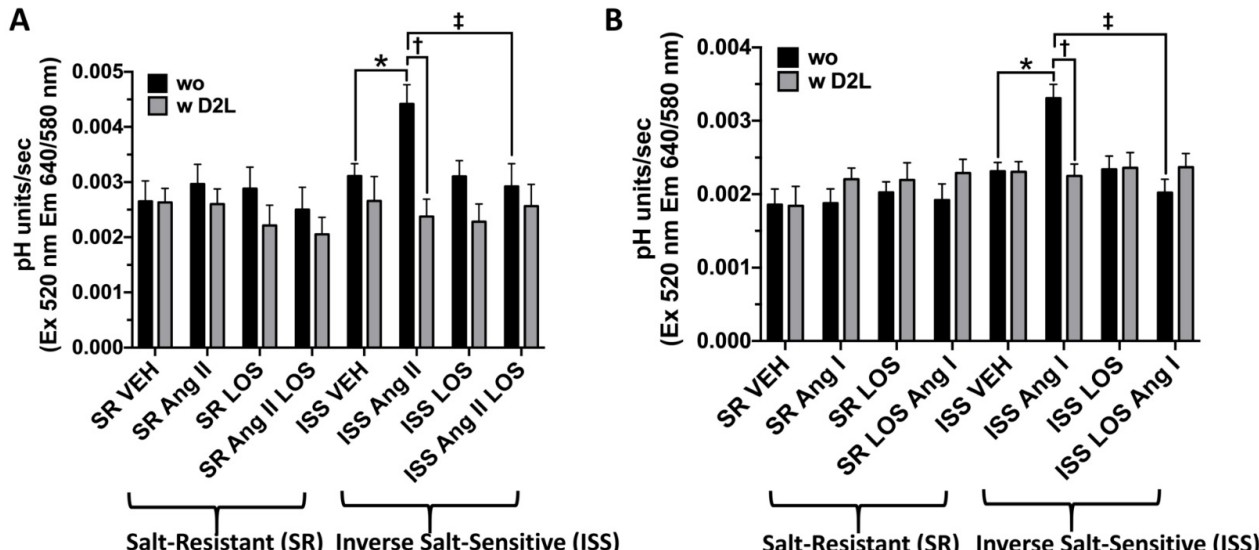

**Figure 6.** pH recovery in SR and ISS cells under Ang II and Ang I treatments. SR and ISS uRPTCs were plated in 96-well plates incubated in 90 mmol/L NaCl, osmolality adjusted to 290 with mannitol, labeled with SNARF1 pH indicator, acidified with ammonium chloride (20 mmol/L, 10 min), switched to sodium-free media (290 osmolality/mannitol) with or without angiotensin II (Ang II, (**A**)) or angiotensin I (Ang I, (**B**)) (10 nmol/L), for 5 min with or without the addition of the $AT_1R$ antagonist, losartan (LOS, 1 μmol/L, 5 min). pH was measured in a microplate fluorometer using a custom single excitation dual emission filter set every 10 s with pH recovery initiated by low sodium buffer (90 mmol/L Sodium as NaCl) and pH recovery measured as pH units/sec. (**A**) Ang II increases pH recovery in low sodium concentration only in ISS uRPTCs (ISS Ang II $0.0044 \pm 0.0003$ vs. ISS VEH $0.0031 \pm 0.0002$, $n = 6$/group, Two-way ANOVA, Tukey's test, * $p < 0.05$) and the $AT_1R$ antagonist, losartan, LOS completely blocks the response (ISS Ang II $0.0044 \pm 0.0003$ vs. ISS Ang II LOS $0.0029 \pm 0.0004$, $n = 6$/group, Two-way ANOVA, Tukey's test, ‡ $p < 0.05$). Stable transfection of *DRD2* (wD2L) completely blocks the low sodium Ang II-dependent increase in pH in uRPTCs from individuals with ISS (ISS Ang II $0.0044 \pm 0.0003$ vs. ISS Ang II w $D_2R$ $0.0024 \pm 0.0003$, $n = 6$/group, Two-way ANOVA, Holm-Sidak test, † $p < 0.05$). (**B**) The Ang II precursor peptide, Ang I, increases pH recovery in low sodium, only in ISS uRPTCs (ISS Ang I $0.0033 \pm 0.0002$ vs. ISS VEH $0.0023 \pm 0.0001$, $n = 6$/group, Two-way ANOVA, Tukey's test, * $p < 0.05$) and LOS completely blocks the response (ISS Ang I $0.0033 \pm 0.0002$ vs. ISS Ang I LOS $0.002 \pm 0.0002$, $n = 6$/group, Two-way ANOVA, Tukey's test, ‡ $p < 0.05$). Stable transfection of *DRD2* (wD2L) completely blocks the low sodium Ang I-dependent increase in pH in uRPTCs from individuals with ISS (ISS Ang I $0.0033 \pm 0.0002$ vs. ISS Ang I w $D_2R$ $0.0022 \pm 0.0002$, $n = 6$/group, Two-way ANOVA, Holm-Sidak test, † $p < 0.05$). wo (non *DRD2*-transfected uRPTCs), wD2L (*DRD2* Long-transfected uRPTCs). ISS = inverse salt-sensitive; SR = salt-resistant.

The Ang II precursor peptide, Ang I, increased pH recovery in low salt concentration only in ISS uRPTCs without *DRD2long* (wo) overexpression (ISS Ang I vs. ISS VEH, $n = 5$ per group, * $p < 0.05$), an effect that was completely blocked by the AT$_1$R antagonist LOS (ISS Ang I vs. ISS Ang I LOS, $n = 5$ per group, ‡ $p < 0.05$), which by itself had no effect (Figure 6B). Stable overexpression of *DRD2 long* (w D2L) completely blocked the low salt Ang I-dependent increase in pH in ISS uRPTCs (ISS Ang I vs. ISS Ang I w D2L, $n = 5$ per group, † $p < 0.05$) (Figure 6B).

### 3.9. RPTC Plasma Membrane Amino Peptidase N Expression

Amino peptidases A (APA) and N (APN) convert Ang II to Ang III and Ang III to Ang IV, respectively, with Ang III abrogating the effects of Ang II [46,50,82]. Plasma membrane APN expression was measured in uRPTCs from individuals with ISS or those with SR in either normal salt (140 mM sodium as NaCl) (NS) containing media or iso-osmotic media (290 mOsm/L) containing 90 mM sodium as NaCl (LS), as above. In-cell immunofluorescence staining in 96 well plates using an extracellular epitope-specific antibody to APN showed that plasma membrane APN expression was greater in ISS than SR uRPTCs. APN expression ($60.20 \pm 6.95\%$) was higher in ISS than SR uRPTCs incubated in 140 mmol/L sodium as NaCl ($p < 0.01$, ISS vs. SR, $n = 18$/group) and increased further to $72.71 \pm 9.83\%$ with incubation in 90 mmol/L sodium ($p < 0.05$, ISS in 90 mmol/L sodium as NaCl vs. ISS 140 mmol/L sodium as NaCl, $n = 18$/group) (Figure 7).

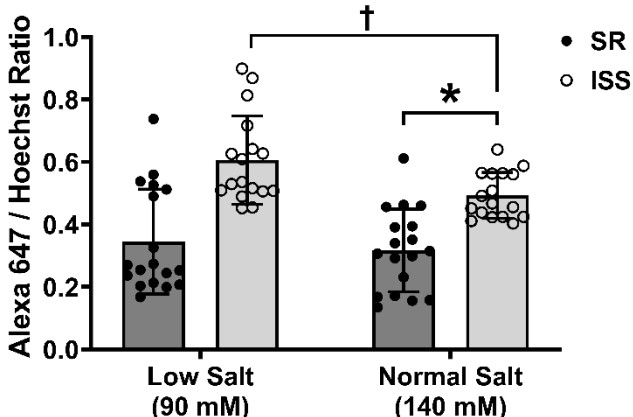

**Figure 7.** Plasma membrane Aminopeptidase N (APN) expression in SR and ISS cells under low salt treatment. In-Cell immunofluorescence staining in 96 well plates using an extracellular epitope specific antibody to APN shows that ISS cells have a higher plasma membrane APN expression than SR cells (Normal salt: ISS $0.49 \pm 0.02$ vs. SR $0.32 \pm 0.03$, * $p < 0.01$, $n = 18$/group, Two-way ANOVA, Tukey's test). ISS uRPTCs have a further increase in APN expression when cultured under low salt concentration vs. the same cells cultured under normal salt concentration (ISS low salt $0.61 \pm 0.03$ vs. ISS normal salt $0.49 \pm 0.02$ † $p < 0.05$, $n = 18$/group, Two-way ANOVA, Holm-Sidak post hoc test). ISS = inverse salt-sensitive; SR = salt-resistant.

### 3.10. D2 SNPs and SLC5A11 SNP

The number of SNPs in *DRD2* (rs6276, rs6277) correlates with the increase in MAP [57]. We also found that a *SLC5A11* SNP, rs11074656, is associated with ISS (Table 1). The odds ratio is 2.29, indicating that the odds of expressing ISS are 2.29 times higher in individuals who have rs11074656 compared with individuals who express the normal homozygous major variant. In subjects carrying both SLC5A11 and *DRD2* SNPs, those expressing more total SNPs had higher increase in MAP on low salt diet, relative to high salt diet, i.e., ΔMAP (Low Salt MAP minus High Salt MAP) (Figure 8A). The ΔMAP was significantly higher in those with 5–6 SNPs than those without or with 1 SNP (0–1: $-2.12 \pm 0.81$, $n = 58$; 2–4: $-1.14 \pm 0.52$, $n = 190$; 5–6: $0.98 \pm 0.88$, $n = 54$ $p < 0.05$, one-way ANOVA, Dunnett's test) (Figure 8B). Moreover, there was significantly lower *SLC5A11* expression in those with homozygous SNPs than those with wild-type *SLC5A11* (Figure 8C), possibly contributing

to their elevated BP on low sodium diet, as a result of additional increase in renal sodium reabsorption via the sodium/myo-inositol cotransporter 2 (SLC5A11).

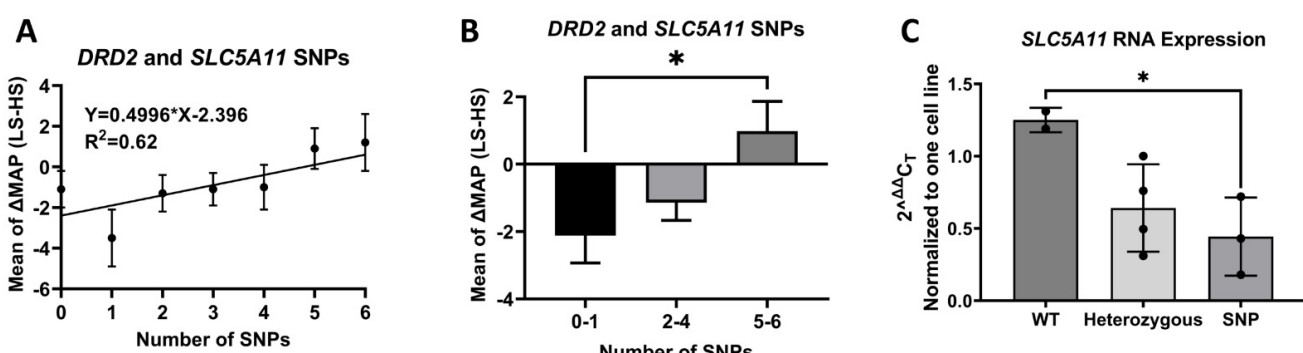

**Figure 8.** Solute Carrier Family 5 Member 11 (SLC5A11) and *DRD2* SNPs. (**A**) There is a linear relationship between the number of SNPs in *DRD2* (rs6276 and rs6277) and SLC5A11 (rs11074656) and the magnitude of change in MAP (ΔMAP) when transitioning between a high and low salt diet ($Y = 0.4996X - 2.396$, $R^2 = 0.62$, $p < 0.05$). (**B**) The cohort was divided into 3 groups, those with 1 or less, those with 2 to 4 and those with 5–6 *DRD2* and SLC5A11 SNPs. ΔMAP is significantly greater in those with 5–6 SNPs than those with 1 or no SNPs (0–1: $-2.12 \pm 0.81$, $n = 58$; 2–4: $-1.14 \pm 0.52$, $n = 190$; 5–6: $0.98 \pm 0.88$, $n = 54$, * $p < 0.05$, one-way ANOVA, Tukey's test). (**C**) Real time RT-PCR shows a significantly lower SLC5A11 expression in cell lines with SNPs than in cell lines with no SNPs, i.e., wild-type (WT) cell lines (WT, $1.25 \pm 0.06$, $n = 2$; heterozygous, $0.64 \pm 0.15$, $n = 4$; SNP, $0.44 \pm 0.16$, $n = 3$; WT vs. SNP, * $p < 0.05$, one-way ANOVA, Tukey's test).

**Table 1.** Distribution of SLC5A11 rare variant, rs11074656, in different salt sensitivities.

| Salt Sensitivity | WT | Heterozygous | SNP |
|---|---|---|---|
| ISS | 9 (29%) | 19 (61%) | 3 (10%) |
| SR | 111 (51%) | 83 (38%) | 24 (11%) |
| SS | 20 (38%) | 23 (43%) | 10 (19%) |

$X^2 = 9.77$ $p < 0.05$.

## 4. Discussion

Understanding the causes and pathogeneses of hypertension are important because it is one of the most important risk factors for CVD [1–3,5–8,11–15,17–19,83]. Hypertension can be caused or exacerbated by an increase in the intake of salt, in individuals with SS [1–15,17–19,29,36–40]. However, about 11% of individuals have increased BP when the intake of salt is very low (<1.2–3 g/day) [23–25,47,84,85]. In one recent long-term study (median 16 years, IQR 12–17 years [26]) individuals with ISS had a 15-fold increase of end-organ damage and mortality compared with individuals with SR. Whereas several genes have been proposed to cause SS [4,36–41,43,45], little is known about genes that contribute to ISS. We hypothesize that there may be many genetic contributors to chronic diseases such as ISS since no single gene has been shown to be solely responsible for SS of BP [36].

We now report that the $D_2R$ may contribute to the pathogenesis of ISS since the number of variants in the *DRD2* UTR, i.e., rs6276/rs6277 is associated with an increase in BP in individuals fed a low sodium diet. The proposed model of ISS in RPTCs under low sodium condition is depicted in cartoon form (Figure 9). In the left panel is the RPTC of individual with SR, in low sodium conditions. The angiotensin peptides are converted by enzymes found in the apical brush boarder and bind and stimulate the corresponding preferred receptors and lead to the proper regulation of sodium hydrogen exchanger type 3, NHE3 and sodium pump, $Na^+K^+$/ATPase to maintain normal sodium reabsorption, maintain sodium homeostasis, and normal blood pressure. An RPTCs from an individual with ISS and *DRD2* SNPs and *SLC5A11* SNPS is depicted in the right panel. In ISS individuals with

*DRD2* SNPs, the lowered expression of $D_2R$ and loss of negative regulation of $AT_1R$ and APN leads to a loss of Ang III and $AT_2R$ signaling and enhanced $AT_1R$ sensitivity. Increased sensitivity to Ang II leads to increased sodium reabsorption, as well as the downstream effects of reduced NRF2, CTCF, and SOD2. The decreased expression of NRF2 also leads to decreased expression of *SLC5A11* which is further exacerbated by SNPs in *SLC5A11* that also reduce expression.

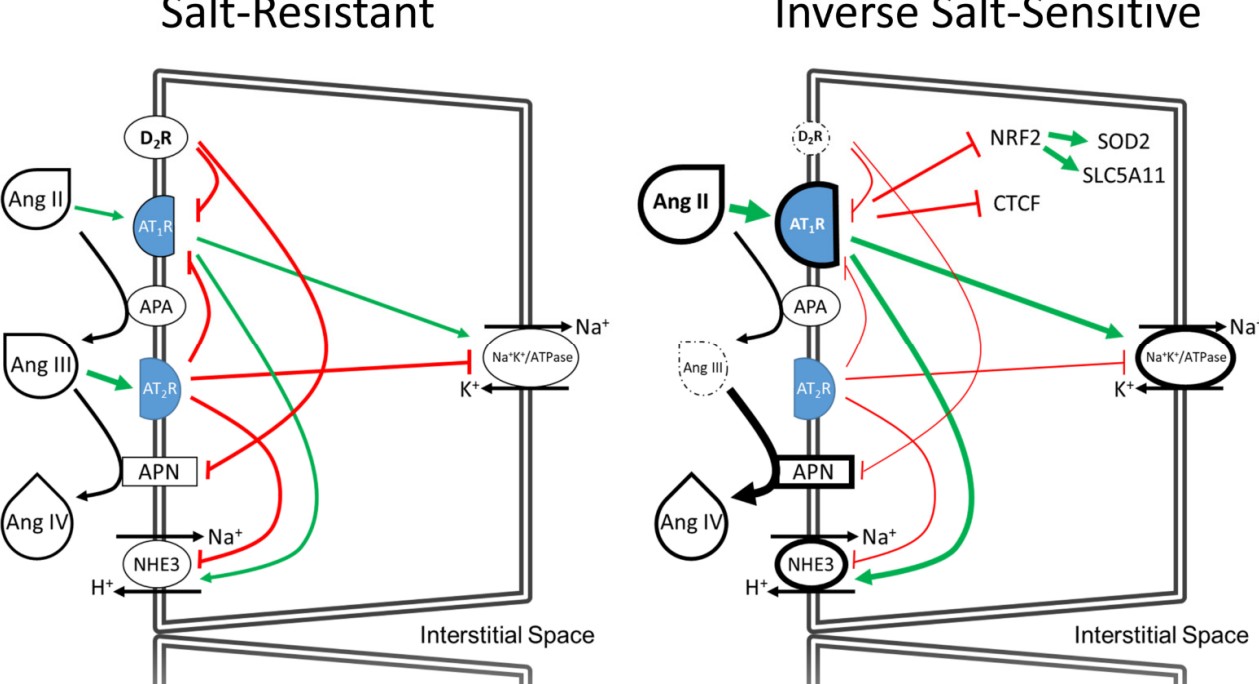

**Figure 9.** A model comparing SR and ISS under low salt condition. In low salt conditions juxta-glomerular cells increase renin and hepatocytes produce more angiotensinogen, delivering increased angiotensin I (Ang I) to the renal proximal tubule where angiotensin converting enzyme, ACE1, on the surface of the proximal tubule cell converts Ang I to Ang II. In ISS, angiotensin type 1 receptor $(AT_1R)$ is sensitized due to the decreased negative regulation by $D_2R$. Increased $AT_1R$ signaling leads to increase sodium transport. ISS in low salt increases the expression of APN even more, causing a reduction of Ang III, the preferred agonist of $AT_2R$, and further sensitizing the $AT_1R$ by reducing the negative regulation of the $AT_2R$ on $AT_1R$ signaling. A feed-forward system is then established by increased $AT_1R$ signaling leading to increased renin, angiotensinogen, ACE1, and lowering APA, in order to raise Ang II and lower Ang III peptide levels. The increased $AT_1R$ signaling lowers NRF2 and SOD2 levels leading to increased free radical damage to the cell, as well as decreased positive epigenetic regulation of *DRD2* via CTCF. Green lines represent positive cell signaling pathways with thicker green lines representing enhanced positive signaling; Red lines represent negative cell signaling pathways and thinner red lines representing decreased negative regulation; Dotted circles around proteins represent decreased protein expression.

*DRD2* variants rs6276 and rs6277 reduce $D_2R$ expression/activity [57], reduced $D_2R$ expression/activity is associated with hypertension and related diseases, such as obesity and metabolic syndrome [86–88]. The long isoform of $D_2R$ is the isoform expressed in the kidney [71]. The $D_2R$ has been shown to inhibit renal sodium transport and contribute to the maintenance of normal fluid and electrolyte balance and BP [54,56,79,80,86,89,90]. $D_2R$ is expressed in RPT and other nephron segments in rats [71,86], mice [91], and humans [92]. $D_2R$ expression in the medullary collecting duct is species-specific [86,93].

One mechanism by which *DRD2* variants rs6276 and rs6277 impair $D_2R$ function is via a decrease in $D_2R$ expression [57,74]. The overall effect of dopamine receptors on renal sodium transport is modulated by other receptors and hormones [90,93], including those

related to the renin-angiotensin-aldosterone system [45,54,94]. The ability of Ang II to stimulate $Na^+$, $K^+$/ATPase activity in rat RPTs is impaired [54], by the $D_2$-like receptor agonist, bromocriptine ($D_2R = D_3R > D_4R$) [54,95]. Our studies show that either Ang II (10 nmol/L) or Ang I (10 nmol/L) increases luminal sodium transport to a greater extent in uRPTCs from individuals with ISS than uRPTCs from individuals with SR. These effects are probably exerted at the $AT_1R$ because they were blocked by the $AT_1R$ blocker, losartan. The apparent increase in $AT_1R$ activity in uRPTCs in individuals that express ISS is due to decreased $D_2R$ expression in ISS, related to *DRD2* variants rs6276 and rs6277, because overexpression of *DRD2* long abrogates the greater increase in luminal sodium transport induced by Ang II in these uRPTCs from individuals with ISS.

APA and APN convert Ang II to Ang III and Ang III to Ang IV, respectively, with the natriuretic effect of Ang III opposing the antinatriuretic effect of Ang II [46,82,96]. Ang II and Ang III can bind to both the $AT_1R$ and the $AT_2R$, but the affinity of Ang II for $AT_1R$ is higher than for $AT_2R$ while the affinity of Ang III for $AT_2R$ is higher than for $AT_1R$. Ang IV binds to the Ang IV Receptor, also known as IRAP and inhibits protease activity [52]. In uRPTCs incubated in normal sodium concentration (140 mM sodium as NaCl), APN expression which is already higher in uRPTCs from individuals with ISS than those with SR is increased to a greater extent by low sodium concentration (90 mM sodium as NaCl) in individuals with ISS than SR. Ang IV has been reported to decrease renal sodium transport [97] and increases sodium excretion in some studies but has no effect on sodium excretion in other studies [98]. Ang IV may have effects on the renal vasculature and renal blood flow that are biphasic with the vasoconstrictor and pressor responses being mediated through $AT_1R$ while the prolonged vasodilator activity being through IRAP [99]. IRAP knockout mice do not have altered blood pressure, but do have increased vasopressin sensitivity during pregnancy [53]. However, we have reported that Ang III increases renal sodium excretion by inhibiting luminal NHE3 and basolateral $Na^+$/$K^+$-ATPase activity in Wistar-Kyoto but not spontaneously hypertensive rats [96]. Thus, the increased activity of Ang II in individuals with ISS can be related to a decreased counter-regulation of Ang III via $AT_2R$. We hypothesize that this is due to decreased stability of Ang III in individuals with ISS in low sodium concentration caused by higher APN expression/activity.

In addition to the contribution of the $D_2R$ in the maintenance of homeostasis by the regulation of renal sodium transport and BP, another protective action of the $D_2R$ via the suppression of reactive oxygen species (ROS) production and inflammation [44,59,60,72,79,80,89–93,100–102]. This occurs, in part, through heterodimerization with receptors and proteins such as DJ1 which prevents the degradation of the antioxidant NRF2 [78]. We have shown that NRF2 is lower in individuals with ISS than individuals with SR under low, but not high, sodium concentration, which could be at least one mechanism why a decrease in the expression of $D_2R$ increases inflammation [57,80]. uRPTCs from ISS individuals have reduced NRF2 expression, as well as reduced expression of several downstream genes of NRF2, such as superoxide dismutase (SOD2), and CTCF [101]. CTCF stimulates the $D_2R$ so that the reduced expression of CTCF in ISS may be part of the mechanism that decreases $D_2R$ expression.

Epigenetics may also be involved in the etiology of ISS. The epigenetic regulation of gene transcription and translation can be influenced by diet [102–105]. For example, an increase or decrease in oxidative stress caused by diet can provoke changes in epigenetics [106,107]. Thus, increased salt intake can increase oxidative stress and oxidative stress can influence epigenetics [108–111]. The renal $D_2R$ and $D_5R$ receptors regulate several redox enzymes [44,59,79,80,89,90,93,111,112]. We have previously reported that miR-124 expression is increased in urinary exosomes of individuals with SS [33]. miR-124 can regulate c-Myc, which we have reported to increase GRK4 expression which impairs $D_1R$ function [113]. C-myc is negatively regulated by CTCF [114], a conserved transcriptional repressor found in RPTCs that binds to a region of the chromosome 100 bases away from the *DRD2* variant rs6276. We, then, studied miR-485-5p, which lies in the 3″ UTR of the *DRD2*. miR-9-5p, miR-485-5p, and miR-137 regulate schizophrenia risk genes and miR-9-5p targets the *DRD2*, a drug-target in schizophrenia [74–76]. In this report, we demonstrated

that miR-485-5p only binds to the rs6276 $D_2$R SNP, thus reducing $D_2$R expression. We used a miRNA mimic that behaves just like the endogenous mir-485-5p miRNA and found it to reduce the expression of $D_2$R, but only in the uRPTCs obtained from individuals with ISS. When transfected with a miRNA blocker, the endogenous miRNA is inhibited and the expression of $D_2$R returns to normal. We have reported 14 additional micro RNAs to be associated with ISS [33]. Among the 14 studied miRNAs is miRNA 30c-1-3p which is pro-inflammatory [115]; its role in the inflammation that occurs with decreased $D_2$R expression [44,57,59,60,89,93,100–102] remains to be determined.

SNPs involved with aberrant blood pressure regulation, especially those involved with sodium balance, could contribute to either the etiology of ISS or SS when compared to SR. Ji and others have discovered SNPs that confer protection against hypertension [116]. We recently showed that one subunit of the $Na^+$ channel (alpha ENaC) is involved in the etiology of ISS [117]. Here, we find that the RPT apical $Na^+$/myoinositol cotransporter 2 (aka SMIT2, *SLC5A11*, and *SGLT6*) containing homozygous SNP at rs11074656 is decreased in individuals with ISS [47]. In addition to the SNP-dependent decrease in expression of SLC5A11, the reduced expression of SLC5A11 in low salt conditions in ISS may be due to SLC5A11 being a known positively regulated specific target of the transcription factor, NRF2 [118]. Similarly, the increased expression of SLC5A11 in high salt conditions may be related to our previously published pathway of high salt induction of HNF4A in hRPTCs [72], and HNF4A enhances the expression of SLC5A11 [119].

## 5. Conclusions

We demonstrated that *DRD2* variants rs6276 and rs6277, which decrease $D_2$R expression and function, may be important in pathogenesis of ISS, a state in which BP is increased by a low salt diet. The number of *DRD2* rs6276/rs6277 is associated with an increase in BP when sodium intake is changed from high salt (300 mmol sodium/day) diet to a low salt diet (10 mmol sodium/day). The presence of *DRD2* rs6276/rs6277 decreases the RPTC plasma membrane expression $D_2$R and its affinity to $D_2$-like receptor antagonist. The decreased expression of $D_2$R in cells expressing *DRD2* rs6276/rs6277 can be related to increased binding of miR-485-5p miRNA at rs6276. The presence of *DRD2* rs6276/rs6277, by themselves, does not affect uRPTC luminal sodium transport, but is augmented by Ang II, via the $AT_1$R. The expressions of the antioxidant proteins NRF2 and SOD2 are decreased in uRPTCs with *DRD2* rs6276/rs6277 incubated in low (90 mmol/L NaCl) but not normal (140 mmol/L NaCl) or high salt (190 mmol/L NaCl) concentration. We conclude that the elevated BP found in individuals with ISS in response to a low salt intake could be caused by polymorphisms in *DRD2* that decrease $D_2$R expression and function, resulting in increased luminal sodium transport in the RPT caused by increased sensitivity to Ang II, via $AT_1$R. A decreased concentration of Ang III caused by an increase in APN expression may also play a role in the ISS phenotype. We also found another SNP, *SLC5A11* rs11074656, that is also positively associated with ISS. Given that ISS and SS constitute ~33% of the US population and both are associated with increased morbidity and mortality [110,111] then there is a need for diagnostic markers for these conditions. Surrogate markers obtained from random urine specimens may be used in the future to determine each individual's personal salt index [20,26,33,40,44] that might encourage dietary adjustments leading to improved population health.

**Author Contributions:** Conceptualization, J.J.G. and R.A.F.; methodology, J.J.G.; validation, J.J.G., R.M.C., P.A.J. and R.A.F.; formal analysis, J.J.G., P.X. and K.A.S.; investigation, J.J.G. and P.X.; resources, R.A.F.; data curation, J.J.G. and P.X.; writing—original draft preparation, J.J.G., P.X. and R.A.F.; writing—review and editing, P.X., K.A.S., W.Y., R.M.C. and P.A.J.; visualization, J.J.G. and P.X.; supervision, R.A.F.; project administration, R.A.F.; funding acquisition, R.A.F. All authors have read and agreed to the published version of the manuscript.

**Funding:** This work was supported by the National Heart, Lung, and Blood Institute (NHLBI) P01 HL074940. Carey is Principal Investigator of R01 HL128189 and Project Director for Program Project Grant (P01 HL074940). Jose is Principal Investigator of R01 DK039308 and R01 DK119652 and Project Director of P01 HL074940.

**Institutional Review Board Statement:** The study was conducted in accordance with the Declaration of Helsinki, and approved by the Institutional Review Board of University of Virginia (protocol code HSR# 13310, 8/24/23 and HSR # 11494, 17 June 2021).

**Informed Consent Statement:** Informed consent was obtained from all subjects involved in the study.

**Data Availability Statement:** Not applicable.

**Acknowledgments:** We thank Stephen Marshall, Chase Carson, and Qing Zhang for their support and helpful discussions during the performance of these studies.

**Conflicts of Interest:** The authors declare no conflict of interest.

## Abbreviations

| | |
|---|---|
| Ang I | angiotensin I |
| Ang II | angiotensin II |
| Ang III | angiotensin III |
| Ang IV | angiotensin IV |
| APN | amino peptidase N |
| $AT_1R$ | angiotensin II type 1 receptor |
| $AT_2R$ | angiotensin II type 2 receptor |
| BP | blood pressure |
| $D_2R$ | dopamine type 2 receptor |
| *DRD2* | dopamine type 2 receptor gene |
| GRK4 | G protein-coupled receptor kinase type 4 |
| ISS | inverse salt sensitivity |
| MAP | mean arterial pressure |
| RAS | renin-angiotensin system |
| RPTC | renal proximal tubule cell |
| SR | Salt resistance |
| SS | Salt sensitivity |
| uRPTC | urine-derived renal proximal tubule cell |

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
