# Peer review of "Inverse Salt Sensitivity of Blood Pressure Is Associated with an Increased Renin-Angiotensin System Activity"

_biomedicines, doi:10.3390/biomedicines10112811_

Round 1

Reviewer 1 Report

The authors describe a laborious work with enormous implication that unravels many of the mechanisms involved in the RAS pathway along with the regulation of D2R as a key piece in ISS patients on low-salt diets, for which I express my congratulations.

11)      Please, the authors must define all the abbreviations of the abstract..

22)      In the introduction, the authors detail: “Other components of the renin-angiotensin system (RAS) may also play a role in the pathogenesis of ISS, such as aminopeptidase N (APN) which converts Ang III into angiotensin IV (Ang IV), reducing levels of Ang III, the preferred endogenous agonist for the natriuretic Ang II type 2 receptor (AT2R), resulting in sodium retention and elevated BP.46,47 APN has variable effects on BP.48-50 Decreased cerebral APN activity increases Ang III levels and BP; the cerebral and paraventricular nucleus infusion of APN decreases BP.49 However in the kidney, a defect in Ang III signaling through the AT2R occurs in both prehyper tensive and hypertensive spontaneously hypertensive rats (SHR);51 increased production of Ang III and decreased renal APN activity rescues this defect and inhibits renal sodium transport and decreases BP in SHR.50 By contrast, nanomolar concentrations of Ang IV has been reported to increase BP in rats by interacting with the AT1R.48 The dopamine type 2 receptor (D2R) negatively interacts with the AT1R in many tissues, including the kidney.52,53 DRD2 rs6276 (1347G>A) at 3’UTR (MAF 0.484) has decreased D2R function and DRD2 rs6277 (957C>T exon7 synonymous mutation, MAF= 0.273) has decreased DRD2 mRNA stability.54 D2R has a protective role against inflammation51 and these DRD2 variants increase inflammation and fibrosis in human and mouse RPTCs.55-58 The present studies tested the hypothesis that DRD2 variant-mediated dysregulation of the RAS contributes to the pathogenesis of ISS. Other genes that are expressed in the kidney, such as SMIT1 (SLC5A3)59 and SMIT2 (SLC5A11, SGLT6) (Na+/myoinositol cotransporters), may be related to ISS. SMIT2 is responsible for the apical sodium/myoinositol transport in the RPT.60

Further on page 15 the authors write: “APA and APN convert Ang II to Ang III and Ang III to Ang IV respectively, with the natriuretic effect of Ang III opposing the antinatriuretic effect of Ang II.46,80,94 In uRPTCs incubated in normal sodium concentration (140 mM sodium as NaCl), APN expresión which is already higher in uRPTCs from individuals with ISS than those with SR is increased to a greater extent by low sodium concentration (90 mM sodium as NaCl) in ISS than SR individuals………………………………….. Thus, the increased activity of Ang II in individuals with ISS can be related to a decreased counter-regulation of Ang III via AT2R; we hypothesize that this is due to decreased stability of Ang III in individuals with ISS in low sodium concentration caused by higher APN expression/activity”.

I have serious discrepancies on the subject. Both Ang II and Ang III bind to AT1 and AT2, but AngII with more affinity to AT1 and Ang III to AT2. Ang III is a vasodilator, its greatest effect being by binding to AT2.

Ang IV is not a vasopressor as documented, it has a potent vasodilator action through its binding to the IRAP receptor (aka, CysAP). In fact, it increases renal flow.

References:

DOI: 10.1016/s0196-9781(03)00121-9

DOI: 10.1016/s0167-0115(01)00288-9

DOI: 10.1016/s0167-0115(02)00035-6

Furthermore, these results suggest a role for AlaAP (aka, APN) in renovascular hypertension. I completely agree with the results.

33)      Referring to Figure 7. Immunostaining for APN expression shows that ISS cells have higher plasma membrane APN expression than SR cells.

Aas suggestions. Have you checked by ultracentrifugation to separate the cytosolic and membrane fractions to measure them separately by immunoblotting (for example)?

Secondly. As I said, I agree with the result. The authors have measured the expression, but it would have been interesting to know the activity of the APN and the conversion (measurement) of Ang III to Ang IV. It may be that overexpression with low salt in ISS individuals is an adaptive need for low APN activity, in addition to being saturated with normal (or high, in this case due to APN deficiency) levels of Ang III.

44)      Section 2.1.: The authors indicate the reference of Felder et al., 2022 in APA nomenclature, not Vancouver.

55)      I greatly appreciate figure 9. It excellently summarizes his results and clarifies many details.

Author Response

The authors describe a laborious work with enormous implication that unravels many of the mechanisms involved in the RAS pathway along with the regulation of D2R as a key piece in ISS patients on low-salt diets, for which I express my congratulations.

11)      Please, the authors must define all the abbreviations of the abstract.

        Thank you for pointing this out. We have added full names in the abstract.

22)  In the introduction, the authors detail: “Other components of the renin-angiotensin system (RAS) may also play a role in the pathogenesis of ISS, such as aminopeptidase N (APN) which converts Ang III into angiotensin IV (Ang IV), reducing levels of Ang III, the preferred endogenous agonist for the natriuretic Ang II type 2 receptor (AT2R), resulting in sodium retention and elevated BP.46,47 APN has variable effects on BP.48-50 Decreased cerebral APN activity increases Ang III levels and BP; the cerebral and paraventricular nucleus infusion of APN decreases BP.49 However in the kidney, a defect in Ang III signaling through the AT2R occurs in both prehyper tensive and hypertensive spontaneously hypertensive rats (SHR);51 increased production of Ang III and decreased renal APN activity rescues this defect and inhibits renal sodium transport and decreases BP in SHR.50 By contrast, nanomolar concentrations of Ang IV has been reported to increase BP in rats by interacting with the AT1R.48 The dopamine type 2 receptor (D2R) negatively interacts with the AT1R in many tissues, including the kidney.52,53 DRD2 rs6276 (1347G>A) at 3’UTR (MAF 0.484) has decreased D2R function and DRD2 rs6277 (957C>T exon7 synonymous mutation, MAF= 0.273) has decreased DRD2 mRNA stability.54 D2R has a protective role against inflammation51 and these DRD2 variants increase inflammation and fibrosis in human and mouse RPTCs.55-58 The present studies tested the hypothesis that DRD2 variant-mediated dysregulation of the RAS contributes to the pathogenesis of ISS. Other genes that are expressed in the kidney, such as SMIT1 (SLC5A3)59 and SMIT2 (SLC5A11, SGLT6) (Na+/myoinositol cotransporters), may be related to ISS. SMIT2 is responsible for the apical sodium/myoinositol transport in the RPT.60

Further on page 15 the authors write: “APA and APN convert Ang II to Ang III and Ang III to Ang IV respectively, with the natriuretic effect of Ang III opposing the antinatriuretic effect of Ang II.46,80,94 In uRPTCs incubated in normal sodium concentration (140 mM sodium as NaCl), APN expresión which is already higher in uRPTCs from individuals with ISS than those with SR is increased to a greater extent by low sodium concentration (90 mM sodium as NaCl) in ISS than SR individuals………………………………….. Thus, the increased activity of Ang II in individuals with ISS can be related to a decreased counter-regulation of Ang III via AT2R; we hypothesize that this is due to decreased stability of Ang III in individuals with ISS in low sodium concentration caused by higher APN expression/activity”.

I have serious discrepancies on the subject. Both Ang II and Ang III bind to AT1 and AT2, but AngII with more affinity to AT1 and Ang III to AT2. Ang III is a vasodilator, its greatest effect being by binding to AT2.

Ang IV is not a vasopressor as documented, it has a potent vasodilator action through its binding to the IRAP receptor (aka, CysAP). In fact, it increases renal flow.

Thank you for your advice that we should clarify the angiotensin peptide affinities and activities relating to AT1R, AT2R and IRAP binding. We have added to the discussion, the fact that AngII and AngIII can bind to both AT1R and AT2R and with their respective affinities and that Ang IV binds to IRAP.

As far as effects of Ang IV on blood pressure and cortical blood flow, we believe we have interpreted the paper cited correctly (ref 48) since IV infusion of Ang IV increased blood pressure and cortical blood flow that was blocked by an AT1R antagonist Losartan.  We were however unaware that this was a controversial topic. So, additionally we have added to the discussion the information that these peptides can also affect vasodilation and vasoconstriction via the vasculature, and that IRAP KO mice do not have a blood pressure phenotype in mice.  As far as we are aware, these issues have not been investigated in humans. This is likely due to the ethical lack of access to kidney vascular tissue from ISS clinical participants.

References:

DOI: 10.1016/s0196-9781(03)00121-9

DOI: 10.1016/s0167-0115(01)00288-9

DOI: 10.1016/s0167-0115(02)00035-6

Furthermore, these results suggest a role for AlaAP (aka, APN) in renovascular hypertension. I completely agree with the results.

Thank you, but as stated above, renovascular studies in humans are not feasible with clinical derived tissue.  We do hope others will study this in rat and mouse model systems.

33)      Referring to Figure 7. Immunostaining for APN expression shows that ISS cells have higher plasma membrane APN expression than SR cells.

You are correct. Thank you.

As suggestions. Have you checked by ultracentrifugation to separate the cytosolic and membrane fractions to measure them separately by immunoblotting (for example)?

We have not, but we measured the expression with and without membrane permeabilization and find that only a small fraction of the protein is cytoplasmic (data not shown).  The intracellular localized expression looks like freshly translated protein and is likely on its way to being exported to the cell surface since it is a type 1 transmembrane protein.  There is no appreciable differential localization between ISS and SR cells.

Secondly. As I said, I agree with the result. The authors have measured the expression, but it would have been interesting to know the activity of the APN and the conversion (measurement) of Ang III to Ang IV. It may be that overexpression with low salt in ISS individuals is an adaptive need for low APN activity, in addition to being saturated with normal (or high, in this case due to APN deficiency) levels of Ang III.

We have measured activity and blocked activity and studied the consequences of this regulatory network. We appreciate your great suggestion, however, we believe this is beyond the scope of this already lengthy initial paper.

44)      Section 2.1.: The authors indicate the reference of Felder et al., 2022 in APA nomenclature, not Vancouver.

Thank you. In section 2.1, reference to the statistical method for making cutoff values mathematically are referenced correctly now.

55)      I greatly appreciate figure 9. It excellently summarizes his results and clarifies many details.

Thank you!

Reviewer 2 Report

The authors indicated that the elevated blood pressure (BP) found in individuals with inverse-salt-sensitivities (ISS) in response to a low salt intake could be caused by polymorphisms in dopamine D2 receptor expression and function, resulting in increased luminal sodium transport in the renal proximal tubule cells caused by the activation of renin-angiotensin system (RAS).  This manuscript is important.  However, there are some concerns in this manuscript. 

(1)  Introduction is redundant.  The authors should plainly indicate what is known and what this study will clarify.

(2)  The authors indicated amino peptidase N (APN) expression levels in page 11, last paragraph (3.9. RPTC Plasma Membrane Amino Peptidase N Expression).  However, it is important to show APA expression levels.  Therefore, the authors should indicate the expression levels of APA in addition to APN.

(3)  I think that the data of fenoldopam (FEN)/ vehicle (VEH) in salt resistance is 0.955 but not 0.055.  Therefore, the authors should correct the data in page 6, last paragraph (3.2. Direct AT2R Plasma Membrane Recruitment in uRPTC).

(4)  I think that “Control miRNA” is not correct in Figure 5A.  The authors should correct “Control miRNA” to “Control siRNA”.

(5)  I think that “individuals with ISS than those with SS” (Page 10, line 15) is not correct.  The authors should correct “SS” to “SR”.

Author Response

The authors indicated that the elevated blood pressure (BP) found in individuals with inverse-salt-sensitivities (ISS) in response to a low salt intake could be caused by polymorphisms in dopamine D2 receptor expression and function, resulting in increased luminal sodium transport in the renal proximal tubule cells caused by the activation of renin-angiotensin system (RAS).  This manuscript is important.  However, there are some concerns in this manuscript. 

1) Introduction is redundant.  The authors should plainly indicate what is known and what this study will clarify.

We have added a section to the introduction to simplify what is known and what this study attempts to clarify.

2) The authors indicated amino peptidase N (APN) expression levels in page 11, last paragraph (3.9. RPTC Plasma Membrane Amino Peptidase N Expression).  However, it is important to show APA expression levels.  Therefore, the authors should indicate the expression levels of APA in addition to APN.

This is a very astute observation, and we agree that this is an important enzyme to study. In this first mechanistic paper investigating proximal tubule cells from ISS participants, we chose to focus on the low sodium aspect of the phenotype connected to the D2R SNP and D2R expression levels. The study of APA is beyond the scope of this paper.

Our follow up paper to this rather lengthy paper will emphasize the of mechanisms related the high sodium aspects of the cellular phenotype of ISS.

3) I think that the data of fenoldopam (FEN)/ vehicle (VEH) in salt resistance is 0.955 but not 0.055.  Therefore, the authors should correct the data in page 6, last paragraph (3.2. Direct AT2R Plasma Membrane Recruitment in uRPTC).

Thank you for identifying this now corrected error.

4)  I think that “Control miRNA” is not correct in Figure 5A.  The authors should correct “Control miRNA” to “Control siRNA”.

Thank you for identifying this corrected error.

5)  I think that “individuals with ISS than those with SS” (Page 10, line 15) is not correct.  The authors should correct “SS” to “SR”.

Thank you for identifying this corrected error.

Round 2

Reviewer 2 Report

Because the authors responded adequately to my concerns, I have no special requests.